# A 3D Estimation Method Using an Omnidirectional Camera and a Spherical Mirror

**Yuya Hiruta [1], Chun Xie [2], Hidehiko Shishido [2] and Itaru Kitahara [2],***

[1] Doctoral Program in Empowerment Informatics, University of Tsukuba, Tsukuba 305-8577, Japan; hiruta.yuya@image.iit.tsukuba.ac.jp

[2] Center for Computational Sciences, University of TsukuTsukuba, Tsukuba 305-8577, Japan; xiechun@ccs.tsukuba.ac.jp (C.X.); shishido@image.iit.tsukuba.ac.jp (H.S.)

*** Correspondence: kitahara@ccs.tsukuba.ac.jp

**Abstract:** As the demand for 3D information continues to grow in various fields, technologies are rapidly being used to acquire such information. Laser-based estimation and multi-view images are popular methods for sensing 3D information, while deep learning techniques are also being developed. However, the former method requires precise sensing equipment or large observation systems, while the latter relies on substantial prior information in the form of extensive learning datasets. Given these limitations, our research aims to develop a method that is independent of learning and makes it possible to capture a wide range of 3D information using a compact device. This paper introduces a novel approach for estimating the 3D information of an observed scene utilizing a monocular image based on a catadioptric imaging system employing an omnidirectional camera and a spherical mirror. By employing a curved mirror, it is possible to capture a large area in a single observation. At the same time, using an omnidirectional camera enables the creation of a simplified imaging system. The proposed method focuses on a spherical or spherical cap-shaped mirror in the scene. It estimates the mirror's position from the captured images, allowing for the estimation of the scene with great flexibility. Simulation evaluations are conducted to validate the characteristics and effectiveness of our proposed method.

**Keywords:** 3D estimation; catadioptric imaging system; omnidirectional camera; spherical mirror; epipolar geometry

## 1. Introduction

The growing demand for 3D information in various fields, such as autonomous driving for understanding the traffic environment or providing a free viewpoint in sports events, has led to the rapid spread of technologies for acquiring and displaying 3D information. These technologies can be categorized into two main types: active sensing, which involves irradiating objects with light for measurement, and passive sensing, which utilizes cameras as light-receiving sensors. Active sensing methods, like LiDAR(Light Detection and Ranging) [1], offer direct 3D information but require additional light-emitting devices. Passive sensing methods, specifically cameras, become necessary to incorporate visual information, such as texture, into the measured 3D information. However, there are challenges related to the scale of observation systems, measurement range, and prior knowledge.

The widely-used Structure from Motion (SfM) technique [2] enables high-precision 3D reconstruction using multiple or moving cameras. However, this approach necessitates large-scale observation equipment and dynamic imaging. Researchers have explored estimating 3D information from single-shot images captured by a single camera to achieve compactness. Nonetheless, this leads to a limited observation range. This limitation can be overcome by utilizing omnidirectional cameras.

Deep learning-based 3D estimation from monocular omnidirectional images [3] has garnered attention. However, estimating 3D information for unknown objects poses

difficulties due to the reliance on prior knowledge, specifically training data. Estimating the 3D shape of objects with insufficient training data, such as scenes containing rare objects, may decrease accuracy.

One method for obtaining alternative viewpoints in monocular images is through mirrors. A catadioptric system, commonly found in telescopes, consists of a mirror and lenses. When a camera is used as a lens in such a system, it is called a catadioptric imaging system. By employing a curved mirror, the catadioptric imaging system captures the light rays reflected on the mirror surface, enabling the observation of a wider scene than conventional cameras. However, most catadioptric imaging systems [4–7] have a fixed positional relationship between the mirror and the camera, limiting the observation range and degree of freedom. Agrawal et al. [8] proposed using multiple curved mirrors to estimate 3D information when the positional relationship in the catadioptric imaging system is not fixed. Thus, these require multiple mirrors [6,8] or imaging systems [5,7], resulting in compactness and single-shot imaging loss. This paper proposes a 3D estimation method for a catadioptric imaging system using a single curved mirror with an unknown 3D position. Thus, prior to estimating the 3D information of the observed scene, our method estimates the 3D position of the mirror by analyzing the mirror image region in the captured image. The objective is to estimate a wide range of 3D information from a monocular image captured using a compact device without relying on training data.

The proposed method is considered effective for dynamic tubular objects because its characteristic is that 3D information is estimated more effectively on the side of the system. For example, it can be applied to the estimation of 3D information from an endoscope, a medical device that takes images inside the body. Sagawa et al. [9] utilized an endoscope with an attachment for omnidirectional observation, allowing for wide-range observation inside the body. With a compact device, our proposed method makes it possible to estimate a broad spectrum of 3D information concerning the body, which undergoes temporal changes, from a single-shot image. In addition, since the positional relationship of the system is estimated from the captured images, the system does not require prior calibration, as is the case with stereo systems that use two camera sets, and is robust to external vibrations and long-term use. In stereo, the positional relationship of the camera sets at the calibration time can deviate, leading to decreased accuracy. In tunnel excavation, a vibration of the drill creates such a situation. Thus, this method is considered to be effective for tunnel excavation.

The contributions of this paper are summarized as follows:

- Proposal of a catadioptric imaging system that combines an omnidirectional camera equipped with a fisheye lens and a single spherical mirror.
- Proposal of a compact 3D information estimation method for the scene shown in Figure 1, which does not require prior knowledge by one-shot imaging.

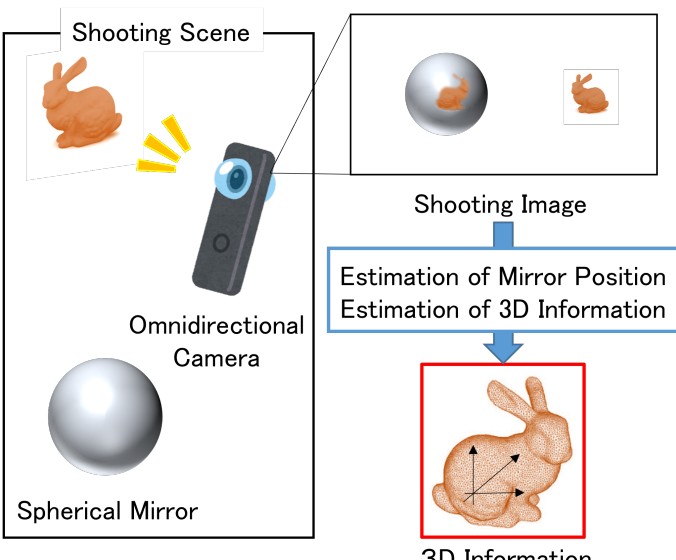

**Figure 1.** Flow of 3D Estimation Using an Omnidirectional Camera and a Spherical Mirror: The capture of two images from different viewpoints using an omnidirectional camera and a spherical mirror, and the estimation of the 3D information of the scene using stereo vision.

## 2. Related Works

### 2.1. Projection Models for the Catadioptric Imaging Systems

The catadioptric imaging systems can be categorized based on the shape of the mirror and the positional relationship between the camera and the mirror. There are two main types of mirror shapes: plane [10–12] and curved [5–8]. Using plane mirrors in imaging systems allows for acquiring 3D information by capturing viewpoints from multiple angles. This approach has a simple projection model but is limited to observing a single object or a relatively narrow scene. A plane mirror of a corresponding size is required to observe wider areas, such as open spaces, resulting in larger imaging equipment. On the other hand, curved mirrors reflect light from various directions, enabling the observation of a wide field of view. Existing methods for acquiring 3D information using catadioptric imaging systems with curved mirrors involve using multiple curved mirrors, employing multiple systems, or moving the systems. However, these methods do not provide a wide range of 3D estimation from monocular images using compact devices.

In this research, we propose a method to obtain 3D information from a single image captured by a catadioptric imaging system comprising a curved mirror and an omnidirectional camera with a fisheye lens. To simplify the analysis of the catadioptric imaging system, we use a spherical mirror with a shape of either a complete sphere or a spherical cap cut in a plane. Analyzing such a system requires information about the incident direction of light rays from an object onto the mirror surface and the position of their reflection on the mirror surface. The projection model varies depending on the positional relationship between the camera and the mirror. When the relationship is fixed [4–6], the projection modeling is relatively straightforward. However, this approach presents challenges, such as the equipment size and accommodating changes in the object's size. On the other hand, when the relationship is variable [8], the projection modeling becomes more complex. However, it allows for observation of the object from any camera position. Agrawal et al. [8] proposed a method for 3D estimation using multiple curved mirrors, which has shown successful results. However, the estimation process requires bundle adjustment with more than three curved mirrors. Therefore, in this research, we adopt a variable method using curved mirrors to enable observation with a high degree of freedom without limiting the observation range in open spaces.

### 2.2. Segmentation of Mirror Image Region

This method requires the estimation of the position of the mirror from the shot image because the positional relationship between the mirror and the camera is not fixed to realize a wide observation range and a high degree of freedom. Therefore, the mirror image region in the shooting image is required to be segmented.

In this method, it is necessary to estimate the position of the mirror from the captured image because the positional relationship between the mirror and the camera is not fixed, allowing for a wide observation range and a high degree of freedom. Consequently, the mirror image region in the captured image needs to be segmented.

Accurately segmenting the mirror image region from a monocular image is generally challenging due to the reflective nature of mirrors. Previous research has explored methods such as using infrared light [13] and photometric stereo [14]. However, these approaches require additional equipment, resulting in larger setups. Alternatively, deep learning methods [15,16] have been employed to address the segmentation challenge. These methods, such as MirrorNet proposed by Yang et al. [15], utilize deep neural networks to detect mirror edges and segment the mirror image region implicitly. By extracting contextual features from multi-layered scale images and comparing these features between the mirror and the surrounding environment, MirrorNet effectively separates the mirror image region from the rest of the image. Importantly, deep learning-based approaches provide a solution that overcomes the limitations of equipment size, making them more practical for implementation in compact imaging systems.

### 3. 3D Estimation Method Using an Omnidirectional Camera and a Spherical Mirror

The proposed 3D estimation process follows the flow shown in Figure 2. In this process, a known spherical mirror is assumed to be placed in the scene, and images of the mirror are captured using an omnidirectional camera.

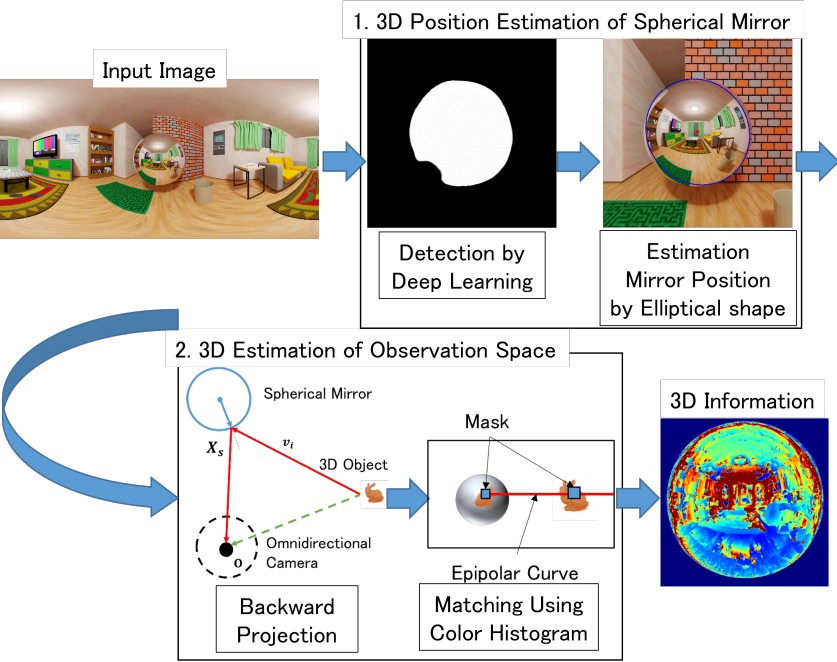

**Figure 2.** The Process of a 3D Estimation Based on a Catadioptric Imaging System with an Omnidirectional Camera and a Spherical Mirror: 1. Segmentation of the mirror image region in an omnidirectional image by network. Estimation of the 3D position of a spherical mirror from the shape of the mirror image (Section 4). 2. Estimation of the 3D information of the shooting scene based on the 3D position of the spherical mirror. Searching for the omnidirectional image part corresponding to the mirror image using the incident light rays from the object obtained by the 3D position information of the spherical mirror, using stereo matching (Section 5).

To estimate the 3D information in the catadioptric imaging system, the determinion of the 3D position of the spherical mirror is crucial. This method estimates the mirror's position by analyzing the mirror image observed in the omnidirectional image. The mirror region is obtained by applying the mirror region segmentation network discussed in Section 2.2 to the omnidirectional image. An ellipse is fitted to the mirror region, and the 3D position of the spherical mirror is estimated based on the shape of the ellipse.

The 3D information of the captured scene is estimated using the estimated 3D position information of the spherical mirror. The process begins with backward projection, which determines the reflection points on the spherical mirror's surface and the incident light ray directions from the objects in the scene. Next, to obtain the 3D information observed from multiple viewpoints, the mirror image at each reflection point and the corresponding area in the omnidirectional image are searched. Stereo matching along the ray directions, estimated through backward projection, is performed to calculate the similarity of visibility. The 3D information with the highest similarity is selected. The similarity of visibility is computed using a color histogram, considering the distortion of the mirror image caused by the spherical shape. This process is applied to all pixels of the spherical mirror in the omnidirectional image.

By following this flow, the proposed method enables the estimation of 3D information in the scene captured by the catadioptric imaging system, leveraging the 3D position estimation of the spherical mirror and the analysis of the mirror reflections.

## 4. 3D Position Estimation of Spherical Mirror

### 4.1. Segmentation of Mirror Image Region in an Omnidirectional Image

To segment the mirror image region in an image, we employ the MirrorNet [15] deep learning method. Given that the model was trained on perspective projection images, we convert the omnidirectional image into six perspective projection images using cube mapping shown in Figure 3. In cases where multiple mirrors are present in the image, we focus solely on processing the mirror located at the center.

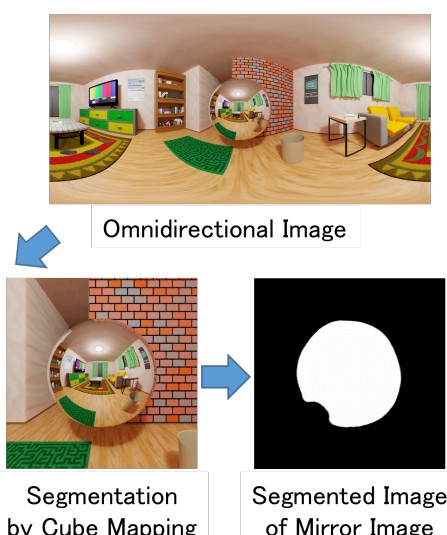

**Figure 3.** Process of segmenting mirror image in omnidirectional image: The procedure begins with the segmenting of the omnidirectional image into perspective projection images via cube mapping. Subsequently, a deep learning model is applied to the perspective projection image containing the mirror image for further segmentation.

### 4.2. 3D Position Estimation Based on Elliptical Shape

The regions segmented using deep learning in Section 4.1 contain errors. In the case of a spherical mirror, the 3D circle is imaged as a circle, whereas for a spherical cap-shaped mirror, the 3D circle of the cut surface is imaged as an ellipse. By applying ellipse fitting, the

shape of the ellipse is ascertained. In conjunction with our prior knowledge of the mirror, such as the radius $R$ of the mirror sphere, the support plane that contains the 3D circle is identified . Consequently, this method makes it possible to determine the 3D position $C_s$ of the mirror.

A diagram of the relationship between the camera and the spherical mirror is shown in Figure 4. In the XYZ coordinate system where the camera position is the origin $O$, the imaging plane is $Z = f$, the horizontal axis on the imaging plane is the x-axis, and the vertical axis is the y-axis. The ellipse on the image plane can be represented as a matrix $\mathbf{Q}$, and the plane supporting the 3D circle in the scene can be estimated by considering the rotation around the camera to the ellipse [17,18]. Using the eigenvalues $\lambda_1$, $\lambda_2$, $\lambda_3 (\lambda_2 \geq \lambda_1 > 0 > \lambda_3)$ of the matrix $\mathbf{Q}$ with the determinant as -1 and corresponding unit eigenvectors $u_1$, $u_2$, $u_3$, the normal unit vector $n$ of the support plane is calculated by Equation (1).

$$n = \mathcal{N}\left[ \sqrt{\lambda_2 - \lambda_1}u_1 + \sqrt{\lambda_1 - \lambda_3}u_3 \right] \tag{1}$$

where $\mathcal{N}[\cdot]$ means normalization.

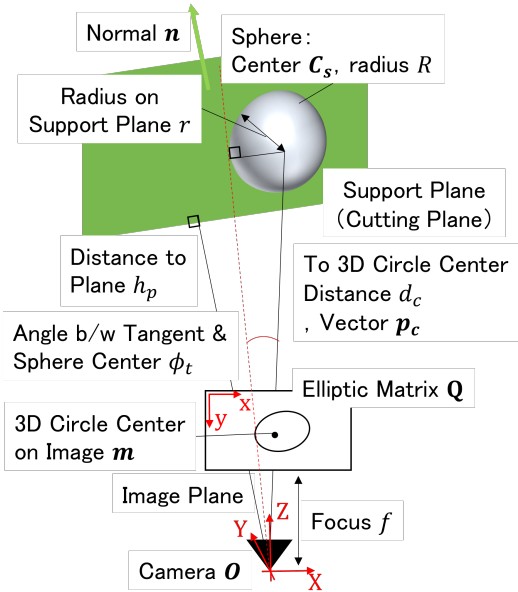

**Figure 4.** Relationship between the Camera and the Spherical Mirror: The estimation of the 3D position of the spherical mirror involves identifying the supporting plane (normal $n$, distance $h_p$ from the camera) using the ellipse shape of the mirror image (matrix $\mathbf{Q}$). This process employs prior knowledge of the spherical mirror's shape, including the radius $R$ of the mirror sphere and the radius r of the 3D circle.

If the radius $r$ of the 3D circle to be imaged is known, the distance $h_p$ from the camera to the support plane $h_p$ is calculated by:

$$h_p = \sqrt{\lambda_1^3}r \tag{2}$$

When the support plane of the 3D circle is perpendicular to the optical axis of the camera, the relationship between the normal $n$ of the plane and the center of the 3D circle on the image $m = (x_C, y_C, f)^T$ is shown in Equation (3). Therefore, the vector $p_c$ from the camera to the center of the 3D circle is calculated by Equation (4), where $\mathbf{K}$ is the camera intrinsic matrix.

$$m = \mathbf{Q}^{-1}n \tag{3}$$

$$p_c = \mathbf{K}m = \mathbf{K}\mathbf{Q}^{-1}n \tag{4}$$

For the support plane of the 3D circle (normal: $\boldsymbol{n}$, distance: $h_p$), let $d_c$ be the length from the camera to the center of the 3D circle, and let $\boldsymbol{p_c}$ be the bearing vector; then, $d_c$ is calculated as:

$$d_c = \frac{h_p}{\boldsymbol{n}^T \boldsymbol{p_c}} \tag{5}$$

The 3D position of the spherical mirror $\boldsymbol{C_s}$ is illustrated in Figure 4 and is calculated by Equation (6)). Substituting $d_c$ and $\boldsymbol{p_c}$ with Equations (4) and (5), we have Equation (7).

$$\boldsymbol{C_s} = d_c \boldsymbol{p_c} + \boldsymbol{n}(\sqrt{R^2 - r^2}) \tag{6}$$

$$= \frac{h_p}{\boldsymbol{n}^T \mathbf{K}\mathbf{Q}^{-1}\boldsymbol{n}} \mathbf{K}\mathbf{Q}^{-1}\boldsymbol{n} + \boldsymbol{n}(\sqrt{R^2 - r^2}) \tag{7}$$

On the other hand, if the radius $r$ of the 3D circle is unknown, the distance $h_p$ from the camera to the support plane cannot be determined using the above method. However, as shown in Figure 5, if a tangent line can be drawn from the camera center to the spherical mirror, it is possible to derive the 3D position of the spherical mirror. Since the sphere is imaged as a circle from any viewpoint and the distance between the point of contact and the sphere center is the sphere radius $R$, the distance to the sphere center is estimated. When the tangent vector to the sphere is obtained, the 3D position $\boldsymbol{C_s}$ of the spherical mirror is obtained by Equation (8). If the tangent line cannot be drawn, the 3D position of the sphere $\boldsymbol{C_s}$ remains indefinite since the actual size of the circle (ellipse) to be imaged is unknown.

$$\|\boldsymbol{C_s}\| = \frac{R}{\sin \phi_t} \tag{8}$$

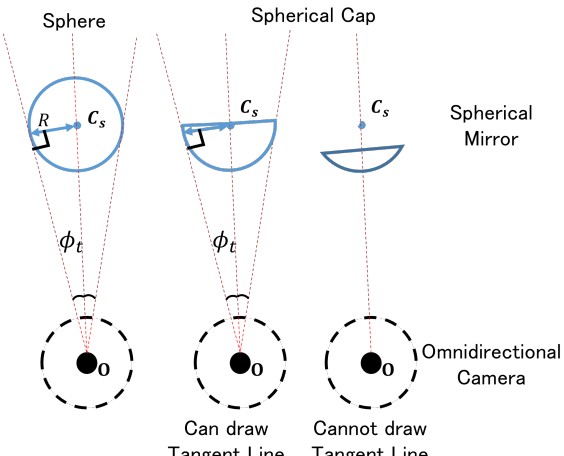

**Figure 5.** The shape of a spherical mirror that allows for the estimation of its 3D position. When the radius of the 3D circle is unknown: On the **left** and in the **middle**, tangent lines can be drawn, allowing for estimation of the mirror's position. On the **right**, tangent lines cannot possibly be drawn, making estimation impossible.

## 5. 3D Estimation of Observation Scene

### 5.1. Lightpass Estimation to the Object Reflected in Mirror Image

Figure 6 shows an overview of the 3D estimation in the observation scene. Here, the reflection point $\boldsymbol{X_s}$ on the spherical mirror and the direction $\boldsymbol{v_i}$ of incident rays from the object to the reflection point on the spherical mirror are obtained by Backward Projection. The equations for the sphere and the reflection point are given by Equations (9) and (10), where $d_s$ is the distance from the camera to the reflection point. The reflected ray $\boldsymbol{v_r}$ from

the spherical mirror to the omnidirectional camera is normalized, and solving for $d_s$ in Equations (9)–(11) can be carried out.

$$\|X_s - C_s\|^2 = R^2 \tag{9}$$

$$X_s = d_s v_r \tag{10}$$

$$d_s = v_r{}^T C_s \pm \sqrt{(v_r{}^T C_s)^2 - (\|C_s\|^2 - R^2)} \tag{11}$$

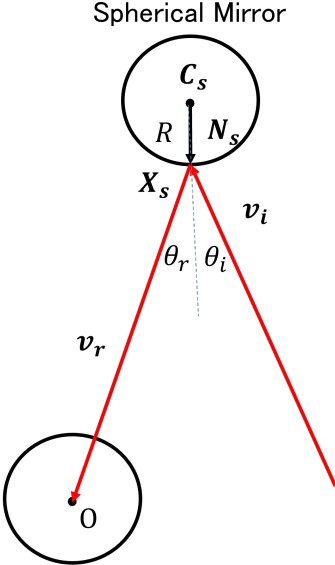

**Figure 6.** Backward Projection (projection from 2D to 3D): Estimation of the light path from an object to the omnidrectional camera via reflection point on the spherical mirror.

After substituting $d_s$ into Equation (10), $X_s$ is derived. Because the incident angle $\theta_i$ and the reflection angle $\theta_r$ are equal with respect to the normal $N_s(\|N_s\| = N_s{}^T N_s = R^2)$ at the reflection point $X_s$ on the mirror surface, the direction of the incident ray $v_i$ from an object to the reflection point on the spherical mirror is determined by Equation (12).

$$v_i = v_r - \frac{2v_r{}^T N_s}{N_s{}^T N_s} N_s = v_r - \frac{2v_r{}^T N_s}{R^2} N_s \tag{12}$$

*5.2. Corresponding Point Search by Color Histogram*

When the incident ray $v_i$ from an object is projected onto the image plane of the omnidirectional camera, the projection image of the ray draws as an epipolar curve. The search process is simplified by rotating the omnidirectional image on the image plane so that the epipolar curve overlaps the center of the vertical axis of the image. The process of the search is shown in Figure 7. Due to the shape of the mirror, the mirror image shot by the camera contains curved surface distortion. As a result, even at the same location, the mirror and omnidirectional images have significantly different appearances. In this paper, a histogram is used, which is less affected by the distortion than the feature that does the corresponding point matching. In order to deal with the distortion, the mask used to obtain the histogram is configured according to the position of the reflection point on the image and the position relationship between the reflection point and the object (i.e. the ratio of the optical path length indicated by the red line and the green line in Figure 7). Additionally, the search range is limited to balance the estimation accuracy and computational cost of the system. The spatial resolution decreases due to the distance between the spherical mirror and the camera. Thus, the search range is terminated when the resolution decreases below

a predefined threshold. The color histogram uses three planes of the HSI color space. The Bhattacharyya distance is used to measure of the similarity of the color histograms.

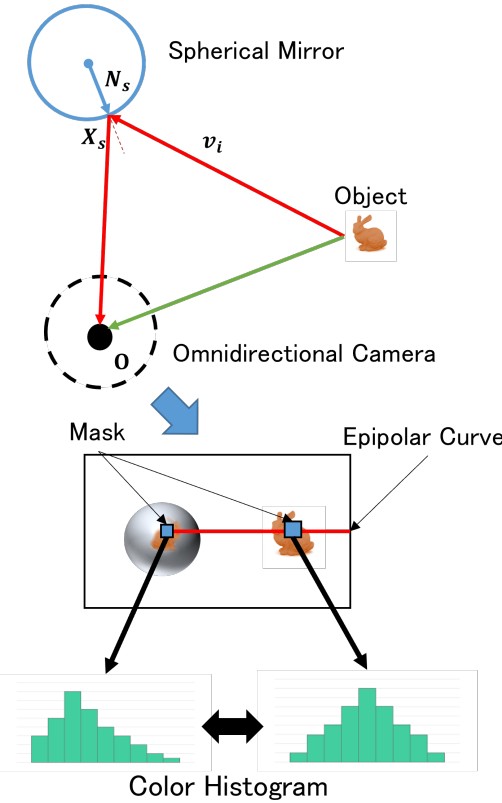

**Figure 7.** Search for Matching Point by Color Histogram: Calculate the color histograms of the mirrored image at the reflection point and the corresponding omnidirectional image. Then, search for corresponding points based on their similarity.

### 5.3. Effect of Errors in 3D Position Estimation of Spherical Mirror on 3D Estimation

Section 4.2 discussed the estimation of the 3D position of the spherical mirror. However, it is crucial to consider the potential errors in this position estimation and their impact on the overall 3D estimation of the object. Figure 8 illustrates this relationship.

In Figure 8, the estimated 3D information is represented by the distance $|P|$ between the camera center and the 3D point. This distance is calculated using Equation (13), which considers the triangle formed by the camera center, the reflection points on the spherical mirror, and the 3D point.

$$
\begin{aligned}
\frac{R}{\sin \phi} &= \frac{d_s}{\sin \theta_s} = \frac{\|C_s\|}{\sin (\pi - \theta_i)} \\
\theta_i &= \arcsin \|C_s\| \frac{\sin \phi}{R} (= \arcsin S) \\
\theta_s &= \arcsin S - \phi \\
d_s &= \|C_s\| \cos \phi - R \sqrt{1 - S^2} \\
\|P\| &= \frac{\|C_s\| \cos \phi - R \sqrt{1 - S^2}}{\frac{1 - 2S^2}{2S \sqrt{1 - S^2}} \sin \theta_n + \cos \theta_n}
\end{aligned}
\tag{13}
$$

Equation (13) allows us to observe the changes in the estimated 3D information resulting from variations in the 3D position of the spherical mirror. In other words, it is

possible to analyze how errors in the estimation of the spherical mirror's position impact the accuracy of the 3D estimation.

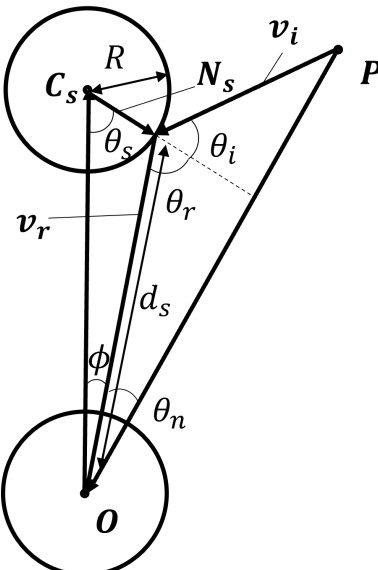

**Figure 8.** Effect of 3D Position Estimation Errors of the Spherical Mirror on 3D Estimation: The effect of the error in the 3D position estimation of the spherical mirror on the 3D estimation is shown by expressing the distance $\|P\|$ from the camera center to the 3D point with the imaging position of the 3D point (the direct image $\theta_i$, the mirror image $\phi$) and the radius $R$ of the spherical mirror.

## 6. Simulation

### 6.1. Simulation with CG Model of a Room

The simulation used a CG model of a room to test the proposed method. A spherical mirror with a 0.5 [m] radius was placed 1 [m] from the camera in the CG environment. The omnidirectional image captured in the experiment and its corresponding ground truth image are shown in Figure 9a,b, respectively. Both images have a resolution of 4096 [pixels] × 2048 [pixels].

The first step of the simulation was to estimate the position of the spherical mirror. The mirror image region in Figure 9a is shown in Figure 10a, and the segmented mirror region is displayed in Figure 10b. An elliptical shape was estimated based on the segmented mirror region, as shown in Figure 10c. The result of the spherical mirror position estimation was 1.03358 [m].

Next, the 3D information estimation results are presented in Figure 11a. The estimation map in Figure 11a visualizes the results by linearly changing the hue, where blue represents the smallest values and red represents the largest values. The difference between the ground truth values and the estimated results at each pixel is shown in Figure 11b. The error map in Figure 11b visualizes the differences by linearly varying the brightness.

The evaluation metric used for assessment is the Mean Absolute Error (MAE), calculated by summing all the pixels' errors and dividing by the total number of pixels. The proposed method was evaluated for all points in the mirror image. In this evaluation, the estimated depth represents the distance between the mirror surface and the camera center for the directly imaged parts of the image that correspond to the mirror surface. At the same time, the ground truth values are the corresponding distances. The error is calculated as the sum of the absolute differences (Sum of Absolute Differences—SAD) between the estimated and the ground truth depths. The MAE value obtained from the proposed method on this evaluation index was 1.15853 [m].

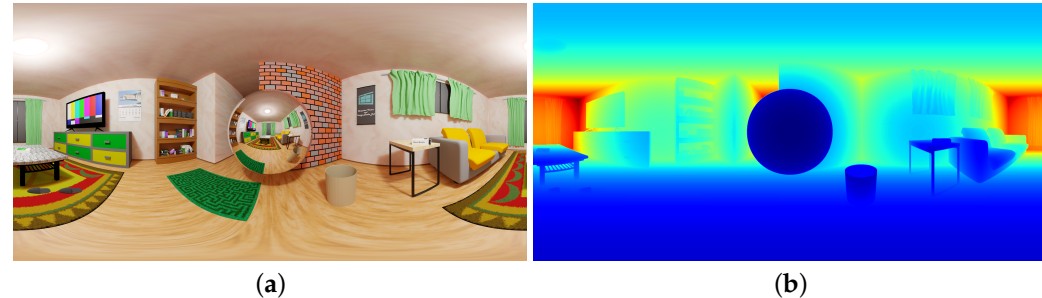

**Figure 9.** Input Images in Comparetion Simulation in Room Model: (**a**) The shooting omnidirectional image. The mirror image is in the center. (**b**) The GT image of the shooting image.

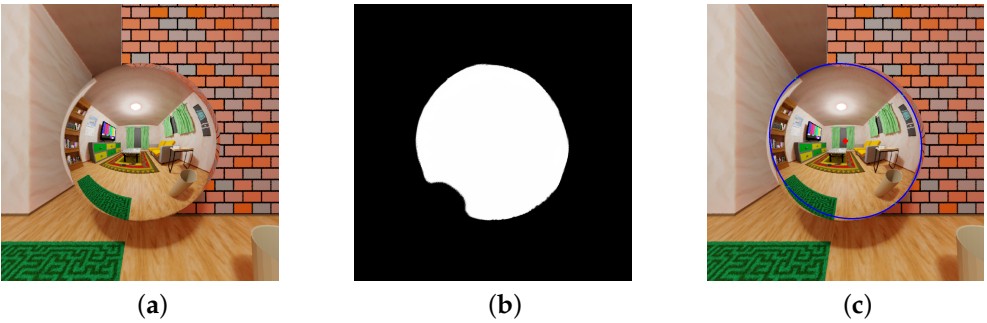

**Figure 10.** Estimation Result Images of Position of Sherical Mirror: (**a**) The image of the mirror image region from the shooting image. (**b**) The image of the segmented mirror region from the image of (**a**). (**c**) The image of the estimated elliptical shape based on (**b**).

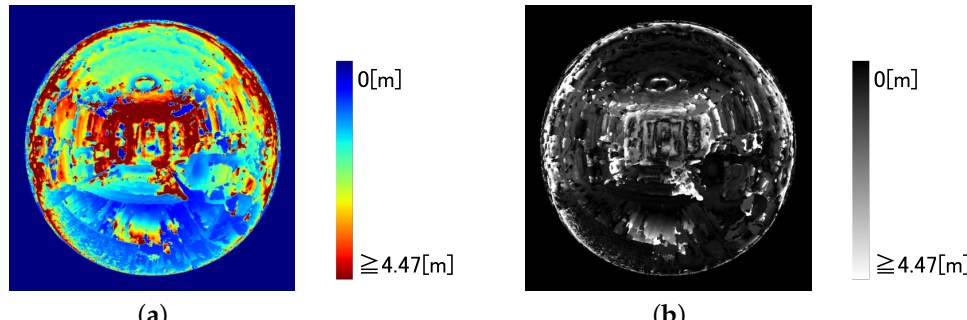

**Figure 11.** Distance and error estimation from the simulated scene: (**a**) The estimation map by the proposed method. The map is visualized by changing the hue linearly. (**b**) The error map. The map is visualized by varying the brightness linearly. The map is obtained by the difference between the ground truth at each pixel and the result estimated by the proposed method. The errors are larger at the edge and the center of the mirror image region.

### 6.2. Discussion of Room Model Simulation

Figure 11b indicates that the proposed method achieves minor errors in many locations within the mirror image, suggesting that estimating the 3D information with reasonable accuracy is possible. However, there are two main reasons for the lower accuracy observed in some cases.

One reason is the difference in appearance between different viewpoints. The direct image captured by the omnidirectional camera and the mirror image reflected by the spherical mirror makes it possible to have different appearances from different positions. These differences correspond to the motion disparity of a camera in stereo, etc. For example, if we focus on the trash can in the lower right corner of the mirror in the figure, its appearance will differ from the trash can reflected in the mirror. As a result, the error increases at the corresponding position in Figure 11b.

Another factor contributing to the lower accuracy is the difference in reflection positions within the mirror image region. Figure 12 illustrates a graph showing the change in estimation accuracy based on the angle between the optical axis and the ray to the camera at the camera center on the spherical mirror. The graph reveals that the Mean Absolute Error (MAE) increases as the angle decreases or increases. These regions correspond to the center and the edges of the specular region. Consequently, these areas in the figure display more prominent errors.

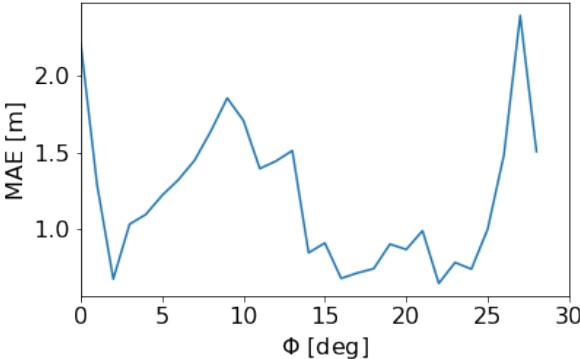

**Figure 12.** Change in Estimation Accuracy against Change in Angle between Optical Axis and Camera Ray in the Spherical Mirror: MAE increases as the angle $\phi$ of incidence at the reflection point $X_s$ decreases or increases. On the other hand, MAE decreases and accuracy is stable in the central area.

To analyze this accuracy discrepancy, a parameter called $h$ is introduced, which represents the size of the line segment perpendicular from the camera to the reflected ray from the object to the reflection point in 3D space (refer to Figure 13). It is defined by Equation (14).

$$h = d_s \|v_i - v_r(v_i{}^T v_r)\| \tag{14}$$

Figure 14 presents the variation in $h$ with the angle between the optical axis and the ray to the camera at the camera center on the spherical mirror. The horizontal axis represents $\phi$ (angle in degrees), and the vertical axis represents $h$ (distance in meters). When comparing this graph with Figure 12, it appears to be inverted. These two figures demonstrate that the estimation accuracy is related to $h$. Smaller values of $h$ indicate more significant changes in the 3D position when shifting one pixel on the omnidirectional image. In other words, the estimated 3D position undergoes significant variations when the matching is shifted by a single pixel, leading to reduced estimation accuracy. As observed in Figure 13, the central and edge regions of the mirror image in the omnidirectional image (the boundary between the omnidirectional image and the non-omnidirectional image part) exhibit small values of $h$. Based on this information, it is possible to assume that the proposed method's accuracy is guaranteed when $h$ is larger than 0.4 m, which corresponds to regions other than the central and edge regions.

The range where the above accuracy is guaranteed is the area of the side of the system due to the characteristics of the optical design. In addition, the accuracy of 3D information estimation decreases as the depth value increases. Considering these characteristics, this method is effective for the 3D measurement of tubular objects. Because of the system's one-shot estimation capability, this method is expected to be effective for endoscopes that capture images of dynamic tubular objects inside the body and for surveying during the excavation of tunnels.

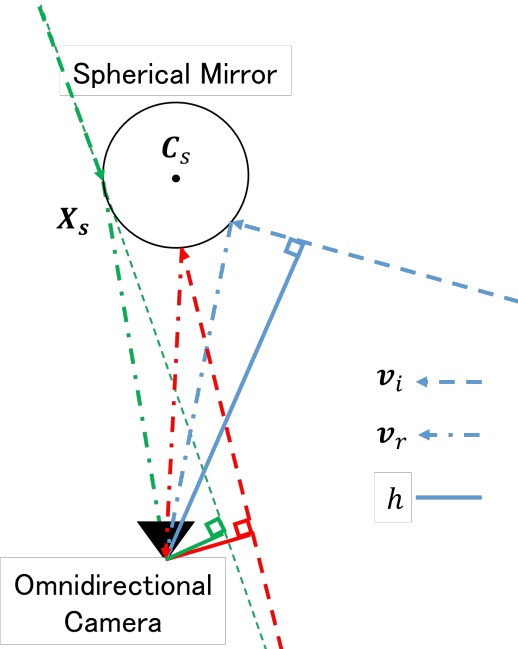

**Figure 13.** Definition of Parameter $h$: The parameter $h$ is defined as the size of the line segment perpendicular from the camera to the incident ray $v_r$ from the object to the reflection point $X_s$ in 3D space.

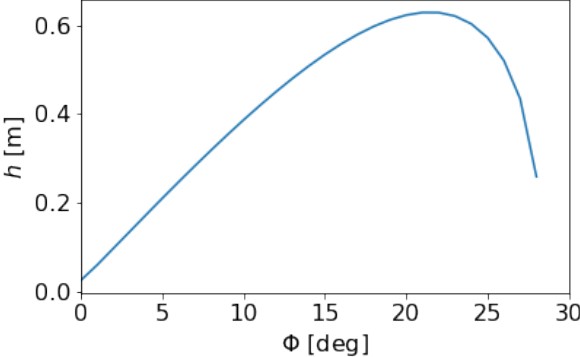

**Figure 14.** Change in Parameter $h$ against Change in Angle $\phi$: The parameters $h$ increase as angle $\phi$ increases or decreases. This feature is similar to the change in MAE against the angle of incidence.

*6.3. Verification of the Effect of 3D Position Estimation Error of a Spherical Mirror on 3D Information Estimation*

The estimation results using the ground truth positions of the spherical mirrors are presented in Figure 15a, while Figure 15b displays the error compared to the ground truth. In this case, the proposed method achieves an MAE of 0.66625 [m]. These estimation results are then applied to the equation that describes the effect of the error in estimating the 3D position of the spherical mirror on the estimation of 3D information, as discussed in Section 5.3. Figure 16 illustrates the outcome of applying Equation to $\|C_s\| = 1.03358$ [m]. It exhibits estimation results similar those of Figure 11a, indicating the potential influence of the estimation of the spherical mirror position.

Furthermore, Figure 16 confirms that the effect of the spherical mirror position estimation is more prominent in the center and at the edge of the mirror image. As discussed earlier, a slight change in the position of the spherical mirror results in a significant variation in the estimated 3D position due to the small value of $h$. Moreover, the reflection position at the mirror's edge differs significantly from that of the spherical mirror, suggesting that the estimated 3D position also undergoes significant changes due to substantial variations in the incident ray from the object.

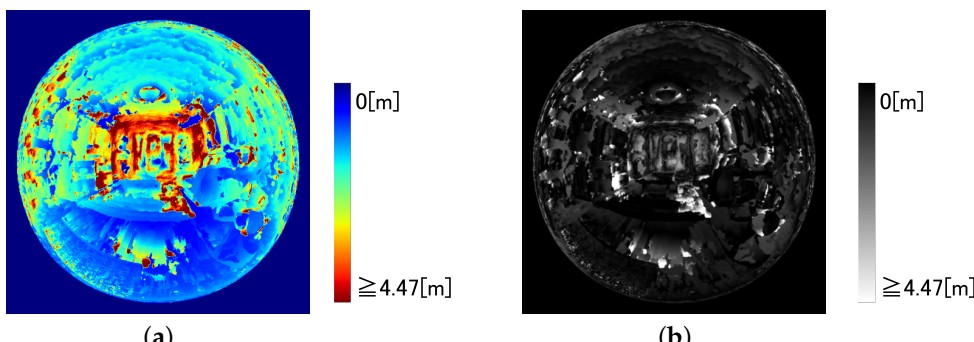

(**a**)　　　　　　　　　　　　　　　　　　　(**b**)

**Figure 15.** Estimation Result Images when Mirror Position is Ground Truth: (**a**) The estimation map using the proposed method, with mirror position as GT. The map is visualized by changing the hue linearly. (**b**) The error map. The map is visualized by varying the brightness linearly. The map is obtained from the difference between the ground truth at each pixel and the result estimated by the proposed method. The overall errors are smaller than when the mirror position is estimated, although the errors are still larger at the edges and in the center of the mirror image.

Next, to examine the relative size of the sphere in the camera's field of view (FOV) and its effect on the distance error, Figure 17 shows the variation in the estimation results for changes in the Z-axis position of the spherical mirror according to Equation (13). As an example of this validation, the poster in the upper right of the mirror in Figure 9a is considered. Figure 17 shows how the estimation of 3D information changes with changes in the position of the spherical mirror when two images are taken at the exact location. The vertical dashed line represents the ground truth of the 3D information, and the horizontal dashed line represents the ground truth of the spherical mirror position.

In Figure 17, the estimated values of the 3D information exhibit sharp changes at a certain point. Its point corresponds to the situation where the imaging position approaches the edge of the mirror image region due to changes in the position of the spherical mirror. If the position of the spherical mirror changes further, it goes beyond the mirror image region, leading to a significant decrease in the reliability of the estimation. On the other hand, when the imaging is conducted in the central region of the mirror image, the changes in the estimated values are smaller. In other words, when the relative size of the sphere in the camera's FOV is small, it becomes more sensitive to changes in the position of the spherical mirror, resulting in a larger impact on distance estimation errors.

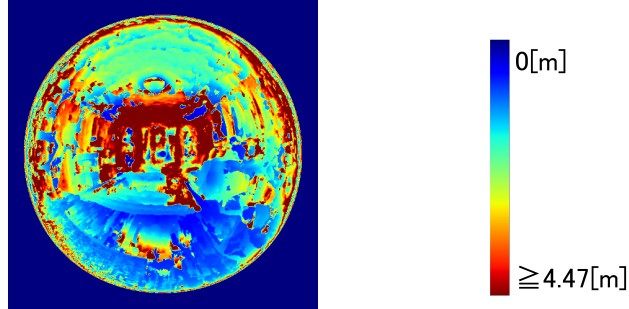

**Figure 16.** Application Result of Equation (13): The result shows that adding the effect of the spherical mirror position estimation to the result in the Figure 11a is similar to the result in Figure 15a.

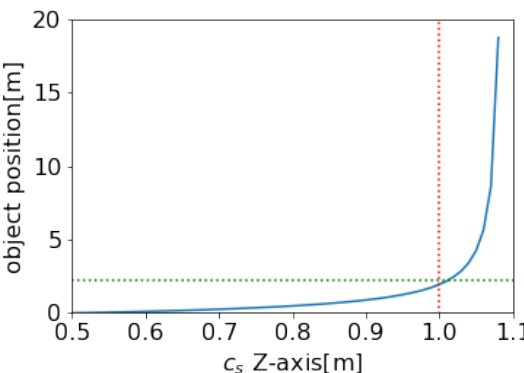

**Figure 17.** Change in 3D Information of the poster against Change in 3D Position of the Spherical Mirror along the Z-axis: Changes in 3D information are indicated by the solid blue line. The green vertical dashed line represents the ground truth of the 3D information, and the red horizontal dashed line represents the ground truth of the spherical mirror position. Since the poster is located at the edge of the mirror surface, if the position of the sphere is estimated to be large in the depth direction, the distance error rapidly increases.

*6.4. Simulation with Image in Real-World*

A simulation of the proposed method is conducted using images captured in real-world scenarios. For the shooting environment, a spherical mirror with a radius of 0.05 [m] is placed at a distance of 0.1[m] in the depth direction and 0.02 [m] in the vertical direction from the camera. The omnidirectional image captured is shown in Figure 18. The image's resolution is 1920 [pixels] × 960 [pixels].

First, the position of the spherical mirror is estimated. The cropped image of the mirror image region is shown in Figure 19a, the segmented image of the mirror image region is shown in Figure 19b, and Figure 19c shows the estimated elliptical shape based on Figure 19b. The result of the position estimation of the spherical mirror was 0.0938 [m] in the depth direction and 0.0252 [m] in the vertical direction.

Next, the estimation result of the 3D information is shown in Figure 20. The estimation map is visualized by changing the hue linearly, where the smallest value is blue, and the largest value is red.

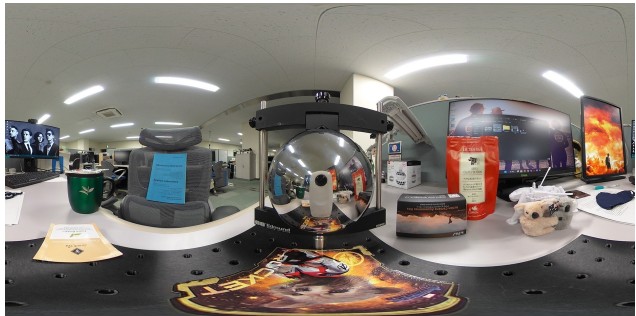

**Figure 18.** Input Images in Real-world Simulation: The shooting omnidirectional image. The mirror image is on the center.

From the estimation results, the 3D estimation of objects placed on the side of the system is achieved. However, it is difficult to estimate the 3D information for objects that are farther away due to the size limitations of the system. Additionally, in the real-world scenario, it is difficult to estimate the 3D image in the area where the camera is reflected in the image.

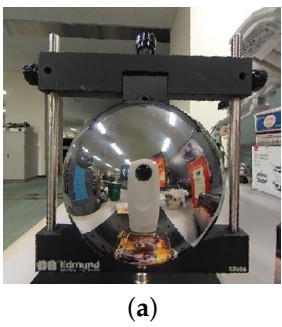
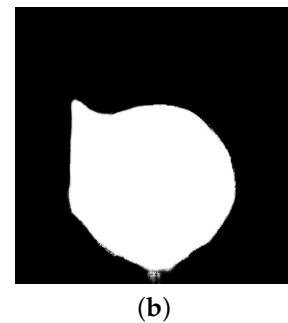
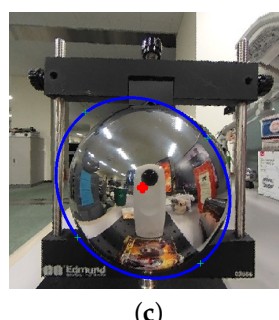

(**a**)　　　　　　　　　　　　　　　(**b**)　　　　　　　　　　　　　　　(**c**)

**Figure 19.** Estimation Result Images of Position of Spherical Mirror: (**a**) The image of the mirror image region from the shooting image. (**b**) The image of the segmented mirror region from the image of (**a**). (**c**) The image of the estimated elliptical shape based on (**b**).

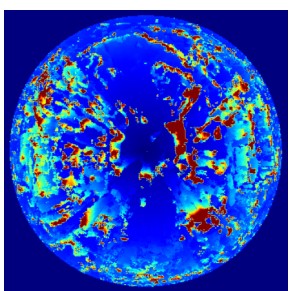
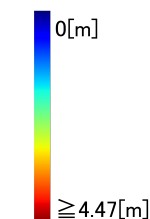

**Figure 20.** Estimation Result Image on Real-World Simulation: The estimation map using the proposed method, with mirror position as GT. The map is visualized by changing the hue linearly.

### 6.5. Limitation

In the position estimation of a spherical mirror, as described in Section 4.2, if the tangent line from the camera center to the spherical mirror cannot be drawn (the mirror shape is spherical, and the mirror appears relatively small due to the positional relationship between the mirror and the camera), the mirror position cannot be estimated, and 3D estimation using this method becomes difficult.

In 3D estimation by the system, the direction connecting the spherical mirror and the camera reduces the resolution and spatial resolution of the estimation or is a dead angle for both, resulting in reduced accuracy and a high possibility that 3D estimation will not be effective. In this paper, a spherical mirror is used to simplify the estimation of the position of the mirror from the captured image. However, the resolution of a spherical mirror degrades at the edge of the mirror image. For this reason, many catadioptric imaging systems use mirrors with hyperbolic or parabolic surfaces. When position estimation is performed on a hyperbolic or parabolic surface, the position estimation becomes more complicated because the shape of the projection onto the camera is not fixed. The mirror's shape is a subject for future research.

### 7. Conclusions

The proposed catadioptric imaging system, which utilizes an omnidirectional camera with a fisheye lens and a spherical mirror, offers a variable positional relationship between the camera and the mirror, allowing for observation with a high degree of freedom. This paper also presents a method for estimating the 3D position of the spherical mirror, enabling 3D estimation from single-shot omnidirectional images without the need for prior learning. The proposed method achieves 3D estimation from compact devices and single image capture to combine the advantages of an omnidirectional camera with a wide observation range and a catadioptric imaging system using curved mirrors. This method eliminates the requirement for prior learning. The effectiveness of the proposed method is demonstrated through simulations using a CG model of a room. The simulations suggest the capabilities and characteristics of the 3D estimation method, proposing potential applications. The paper introduced a novel

catadioptric imaging system and a 3D estimation method that offers flexibility, compactness, and single-shot capability.

**Author Contributions:** Conceptualization, Y.H. and I.K.; methodology, Y.H.; software, Y.H.; validation, Y.H. and I.K.; formal analysis, Y.H.; investigation, Y.H.; resources, I.K.; data curation, I.K.; writing—original draft preparation, Y.H.; writing—review and editing, I.K., H.S. and C.X.; visualization, Y.H.; supervision, I.K.; project administration, Y.H.; funding acquisition, I.K. All authors have read and agreed to the published version of the manuscript.

**Funding:** This work was partially supported by JSPS KAKENHI Grant Number 21KK0070, 22K18303 and 22H01580.

**Institutional Review Board Statement:** Not applicable.

**Informed Consent Statement:** Not applicable.

**Data Availability Statement:** Not applicable.

**Conflicts of Interest:** The authors declare no conflict of interest.

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
