# Peer review of "A 3D Estimation Method Using an Omnidirectional Camera and a Spherical Mirror"

_applsci, doi:10.3390/app13148348_

Round 1

Reviewer 1 Report

1.      The paper needs careful English editing.

2.      In the abstract: “This paper introduces a novel approach for estimating 3D information of a observed scene utilizing” == > should be: “of an observed scene “

3.      There is no or a little discussion which compares between the proposed algorithm and other state-of-the art methods.

4.      Experimentations were done on only one single image, more images need to be utilized to better showcase the methodology.

1.      The paper needs careful English editing.

2.      In the abstract: “This paper introduces a novel approach for estimating 3D information of a observed scene utilizing” == > should be: “of an observed scene “

3.      There is no or a little discussion which compares between the proposed algorithm and other state-of-the art methods.

4.      Experimentations were done on only one single image, more images need to be utilized to better showcase the methodology.

Author Response

R1-1
The paper needs careful English editing.

A. Thank you for your comment. English editing was performed again throughout in the paper. Your comment has helped us to improve the quality of this paper. We surely promise to have the English of this paper, if accepted, checked by native speakers when I write the final manuscript.

R1-2
In the abstract: “This paper introduces a novel approach for estimating 3D information of a observed scene utilizing” == > should be: “of an observed scene “

A. Thank you for your comment. We have made the following corrections to Page 1 Line 7 of this paper. Your comment has helped us to improve the quality of this paper.

R1-3
There is no or a little discussion which compares between the proposed algorithm and other state-of-the art methods.

A. Thank you for your comment. This research targets 3D estimation without prior learning by a one-shot and compact system with a monocular camera, which we consider to be different from the problems targeted by other state-of-the-art methods. Our description was insufficient. We consider the system to be effective in specific situations, as modified and added below to P2 L57 in this paper.

”The proposed method is considered effective for dynamic tubular objects because its characteristic is that 3D information is estimated more effectively on the side of the system. For example, it can be applied to estimating 3D information from an endoscope, a medical device that takes images inside the body. Sagawa et al. [9] have utilized an endoscope with an attachment for omnidirectional observation, allowing for wide-range observation inside the body. With a compact device, our proposed method is possible to estimate a broad spectrum of 3D information concerning the body, which undergoes temporal changes, from a single-shot image. In addition, since the positional relationship of the system is estimated from the captured images, the system does not require prior calibration, as is the case with stereo systems that use two camera sets, and is robust to external vibrations and long-term use. In stereo, the positional relationship of the camera sets at the calibration time can deviate, leading to decreased accuracy. In tunnel excavation, a vibration of the drill creates such a situation. Thus this method is considered to be effective for tunnel excavation.”

R1-4
Experimentations were done on only one single image, more images need to be utilized to better showcase the methodology.

A. Thank you for your comment. We have added the result estimated by the real image in Section 6.4 “Simulation with Image in Real-world” to P14 L338 in this paper.

"6.4 Simulation with Image in Real-world

Simulation of the proposed method is conducted using images captured in real-world scenarios. For the shooting environment, a spherical mirror with a radius of 0.05[m] is placed at a distance of 0.1[m] in the depth direction and 0.02[m] in the vertical direction from the camera. The omnidirectional image captured is shown in Figure 18. The image's resolution is 1920 [pixels] $\times$ 960 [pixels].

First, the position of the spherical mirror is estimated. The cropped image of the mirror image region is shown in Figure 19(a), the segmented image of the mirror image region in Figure 19(b), and Figure 19(c) showing the estimated elliptical shape based on Figure 19(b). The result of the position estimation of the spherical mirror was 0.0938[m] in the depth direction and 0.0252[m] in the vertical direction.

Next, the estimation result of the 3D information is shown in Figure 20. The estimation map is visualized by changing the hue linearly, where the smallest value is blue, and the largest value is red.

From the estimation results, the 3D estimation of objects placed on the side of the system is achieved. However, it is difficult to estimate the 3D information for objects that are farther away due to the size limitations of the system. Additionally, in the real-world scenario, it is difficult to estimate the 3D image in the area where the camera is reflected in the image."

Reviewer 2 Report

Summary:

This manuscript presents a method of 3D image capture by placing a spherical mirror of known diameter in the scene. The distance to the different objects is inferred using the original image from the camera and the reflected image on the sphere surface. An example is provided using computer generated ray tracing scene. An analysis of the distance error is discussed according to the location of the object reflection on the sphere and the sphere calculated distance.

General comments:

The overall methodology and analysis presented in the manuscript is good. The error discussion in interesting. However, the overall interest of the technique is quite limited due to the requirement for a mirror sphere to be physically present in the field of view of the camera. The presence of the sphere will inevitably generate artifacts and obstruct the background view. The claim that the mehtod can be used for endoscopy should include an explanation of how the reflective sphere will be handle.

The manuscript will benefit from a discussion about the relative size of the sphere in the camera FOV and its effect on the distance error,

I also regret that the manuscript does not include a comparison with other techniques such as direct stereoscopy: What if a second omnidirectional camera is used instead of the sphere? Are the processing times to retrieve the distance information similar in both cases (sphere and steropair)? Is there a relationship between the number of pixels dedicated to the distance measurement (secondary image size either with second camera or sphere) and the error?

I would happily reconsider the manuscript if this information is included in the revised version.

Minor revisions:

P1L10: “we are possible” -> it is possible

P1L14: “Experimental evaluations are conducted” -> simulation.

P2L36: “??” -> references missing

P2L44: “the catadioptric imaging system is possible to capture light rays […]” -> the catadioptric imaging system captures the light rays […].

P2L58: “The proposed method is possible to be applied to endoscopes” -> The proposed method can be applied to endoscopes.

P2L71: “The catadioptric imaging systems is possible to be categorized” -> The catadioptric imaging systems can to be categorized.

P3L102: “This method requires estimation of the position” -> This method requires the estimation of the position.

P10L237, L238, L239, L244: “Experiment” -> Simulation (in each of these lines)

P10: figure 9(a) is missing.

P11Figure11 legend: “Estimation Result Images in Comparetion Experiment in Room Model” -> Distance and error estimation from the simulated scene.

P11L269: “The direct image captured by the omnidirectional camera and the mirror image reflected by the spherical mirror is possible to have variations in appearance from different positions” can the authors better explain what they mean by “variation in appearance”?

P12: figure 12 is not clear. The relation between MAE and theta_I is not obvious. Can authors try to use another parameter such as the location on the sphere surface (Phi)?

P14L326: “The effectiveness of the proposed method is demonstrated through experiments using a CG model of a room. The experiments showcase […]” -> simulations.

See general comments

Author Response

R2-1
The claim that the method can be used for endoscopy should include an explanation of how the reflective sphere will be handle

A. Thank you for your comment. The effectiveness of the method proposed in this paper increases on the side of the imaging system consisting of a mirror and a camera, as in the simulation. Therefore, it is not a problem that the mirror prevents the view in the direction the camera is moving, and we assume that the endoscope provides a 3D estimation of the side wall of the tube. Our description was insufficient. We have added the following to P2 L57 in this paper.

"The proposed method is considered effective for dynamic tubular objects because its characteristic is that 3D information is estimated more effectively on the side of the system. For example, it can be applied to estimating 3D information from an endoscope, a medical device that takes images inside the body. Sagawa et al. [9] have utilized an endoscope with an attachment for omnidirectional observation, allowing for wide-range observation inside the body. With a compact device, our proposed method is possible to estimate a broad spectrum of 3D information concerning the body, which undergoes temporal changes, from a single-shot image."

R2-2
The manuscript will benefit from a discussion about the relative size of the sphere in the camera FOV and its effect on the distance error

A. Thank you for your comment. Section  shows the results of estimating the effect of changing the positions of the camera and spherical mirror on the distance error, and the following is added to P14 L321 in this paper.

"Next, to examine the relative size of the sphere in the camera's field of view (FOV) and its effect on the distance error, Figure 17 shows the variation of the estimation results for changes in the Z-axis position of the spherical mirror according to Eq. (\ref{eq:OP2}). As an example of this validation, the poster in the upper right of the mirror in Figure \ref{fig:image}(a) is considered. Figure 17 shows how the estimation of 3D information changes with changes in the position of the spherical mirror when two images are taken at the exact location. The vertical dashed line represents the ground truth of the 3D information, and the horizontal dashed line represents the ground truth of the spherical mirror position.

In Figure 17, the estimated values of the 3D information exhibit sharp changes at a certain point. Its point corresponds to the situation where the imaging position approaches the edge of the mirror image region due to changes in the position of the spherical mirror. If the position of the spherical mirror changes further, it goes beyond the mirror image region, leading to a significant decrease in the reliability of the estimation. On the other hand, when the imaging is done in the central region of the mirror image, the changes in the estimated values are smaller. In other words, when the relative size of the sphere in the camera's FOV is small, it becomes more sensitive to changes in the position of the spherical mirror, resulting in a larger impact on distance estimation errors."

R2-3
I also regret that the manuscript does not include a comparison with other techniques such as direct stereoscopy: What if a second omnidirectional camera is used instead of the sphere? Are the processing times to retrieve the distance information similar in both cases (sphere and steropair)? Is there a relationship between the number of pixels dedicated to the distance measurement (secondary image size either with second camera or sphere) and the error?

A.
Thank you for your comment. When using a stereo pair of omni-directional cameras, the cameras need to be calibrated. However, if the system is subjected to strong vibrations (e.g., tunnel excavation), the prior stereo calibration may not make sense and the estimation accuracy may decrease. On the other hand, the proposed method can estimate the position from the mirror image in the image in a situation. We believe that our method is effective in such situations. Our description was insufficient. We have added the following to P2 L64, P13 L297 in this paper.

"In addition, since the positional relationship of the system is estimated from the captured images, the system does not require prior calibration, as is the case with stereo systems that use two camera sets, and is robust to external vibrations and long-term use. In stereo, the positional relationship of the camera sets at the calibration time can deviate, leading to decreased accuracy. In tunnel excavation, a vibration of the drill creates such a situation. Thus this method is considered to be effective for tunnel excavation."

"The range where the above accuracy is guaranteed is the area of the side of the system due to the characteristics of the optical design. In addition, the accuracy of 3D information estimation decreases as the depth value increases. Considering these characteristics, this method is effective for the 3D measurement of tubular objects. Because of the system's one-shot estimation capability, this method is expected to be effective for endoscopes that capture images of dynamic tubular objects inside the body and for surveying during the excavation of tunnels."

R2 Minor revisions
A. Thank you for your comment. We have made the following corrections to the corresponding parts of the paper. Your comment has helped us to improve the quality of this paper. In addition, we would like to add a special response to the following.

  • P10: figure 9(a) is missing.

A. The image may not be displayed depending on the software you are using because of its large size. “Adobe Acrobat Reader" is able to display the image.

  • P11L269: “The direct image captured by the omnidirectional camera and the mirror image reflected by the spherical mirror is possible to have variations in appearance from different positions” can the authors better explain what they mean by “variation in appearance”?

    A.
    What is meant by "variation in appearance" corresponds to the camera motion parallax in stereo. The proposed system is analogous to a stereo with cameras facing each other, so the motion parallax will be large. Also, "variation" may have caused some misunderstandings. Our description was insufficient. We have made the following corrections and additions.

    “The direct image captured by the omnidirectional camera and the mirror image reflected by the spherical mirror is possible to have difference in appearance from different positions. This corresponds to the motion disparity of a camera in stereo etc. ”
  • P12: figure 12 is not clear. The relation between MAE and theta_I is not obvious. Can authors try to use another parameter such as the location on the sphere surface (Phi)?

A. As you commented, we have replaced Figure12 and 14 expressed in terms of MAE and Phi.

Reviewer 3 Report

This manuscript is interesting. I believe it is appropriate for publication  after some revisions. The first main action which needs to be taken is to explain the details of the method much more clearly. My full list of comments are below.

1. The introduction section must be improved. In particular, the problem of 3D estimation must be presented, as well as a discussion of the state of the art. 

2. The novelty of the paper seems to be the system design, but no justification/benefit of this setup is given.

3. I think the discussion of the experiment is not enough.

4. The whole article focusses on developing all the methods for 3D information. The authors make a strong statement in the conclusion stating that the experiments have demonstrated the effectiveness, but it has not been shown decisively and care should be taken before such statements are made. 

Some spelling and grammatical errors

Author Response

R3-1 The introduction section must be improved. In particular, the problem of 3D estimation must be presented, as well as a discussion of the state of the art.

R3-2 The novelty of the paper seems to be the system design, but no justification/benefit of this setup is given.

A. (R3-1, 2) Thank you for your comments. Our description was insufficient and it was difficult to understand the target 3D estimation problem and the effectiveness of the proposed setup. We believe that the system is effective under certain circumstances, as we modified and added below to P2 L57 in the introduction in this paper.

”The proposed method is considered effective for dynamic tubular objects because its characteristic is that 3D information is estimated more effectively on the side of the system. For example, it can be applied to estimating 3D information from an endoscope, a medical device that takes images inside the body. Sagawa et al. [9] have utilized an endoscope with an attachment for omnidirectional observation, allowing for wide-range observation inside the body. With a compact device, our proposed method is possible to estimate a broad spectrum of 3D information concerning the body, which undergoes temporal changes, from a single-shot image. In addition, since the positional relationship of the system is estimated from the captured images, the system does not require prior calibration, as is the case with stereo systems that use two camera sets, and is robust to external vibrations and long-term use. In stereo, the positional relationship of the camera sets at the calibration time can deviate, leading to decreased accuracy. In tunnel excavation, a vibration of the drill creates such a situation. Thus this method is considered to be effective for tunnel excavation."

R3-3
I think the discussion of the experiment is not enough.

A. Our description was insufficient and the discussion of the simulations in the paper was not sufficient. We have described below the conditions under which the proposed method is valid and a discussion of the limits of adaptation to P2 L64, P15 L357 in this paper.

"The range where the above accuracy is guaranteed is the area of the side of the system due to the characteristics of the optical design. In addition, the accuracy of 3D information estimation decreases as the depth value increases. Considering these characteristics, this method is effective for the 3D measurement of tubular objects. Because of the system's one-shot estimation capability, this method is expected to be effective for endoscopes that capture images of dynamic tubular objects inside the body and for surveying during the excavation of tunnels."

"6.5 limitation

In the position estimation of a spherical mirror, as described in Section, if the tangent line from the camera center to the spherical mirror cannot be drawn (the mirror shape is spherical, and the mirror appears relatively small due to the positional relationship between the mirror and the camera), the mirror position cannot be estimated, and 3D estimation using this method becomes difficult.

In 3D estimation by the system, the direction connecting the spherical mirror and the camera reduces the resolution and spatial resolution of the estimation or is a dead angle for both, resulting in reduced accuracy and a high possibility that 3D estimation will not be effective. In this paper, a spherical mirror is used to simplify the estimation of the position of the mirror from the captured image. However, the resolution of a spherical mirror degrades at the edge of the mirror image. For this reason, many catadioptric imaging systems use mirrors with hyperbolic or parabolic surfaces. When position estimation is performed on a hyperbolic or parabolic surface, the position estimation becomes more complicated because the shape of the projection onto the camera is not fixed. The mirror's shape is a subject for future consideration."

R3-4
The whole article focusses on developing all the methods for 3D information. The authors make a strong statement in the conclusion stating that the experiments have demonstrated the effectiveness, but it has not been shown decisively and care should be taken before such statements are made.

A. Thank you for your comment. We have made the following corrections to P16 L383 of conclusion in the paper. Your comment has helped us to improve the quality of this paper.

“The experiments showcase the capabilities and characteristics of the 3D estimation method, validating its performance and potential applications.”

→“The simulations suggest the capabilities and characteristics of the 3D estimation method, proposing potential applications. ”

Round 2

Reviewer 2 Report

The authors addressed my remarks in their updated manuscript. I would have appreciated a better comparison with the un-calibrated stereoscopic method. However, I can support the revised manuscript for publication in the present form.

The English is good enough

Author Response

Thank you for reviewing our paper.
The quality has been improved thanks to you.